ecology

insecticide resistance, cuticle alterations, legs

**Authors for correspondence:**
Vasileia Balabanidou
e-mail: balaban@imbb.forth.gr
John Vontas
e-mail: vontas@imbb.forth.gr

# Mosquitoes cloak their legs to resist insecticides

Vasileia Balabanidou[1], Mary Kefi[1,2], Michalis Aivaliotis[1,3,4], Venetia Koidou[1,2], Juan R. Girotti[5], Sergio J. Mijailovsky[5], M. Patricia Juárez[5], Eva Papadogiorgaki[2], George Chalepakis[2], Anastasia Kampouraki[1,7], Christoforos Nikolaou[1,2], Hilary Ranson[6] and John Vontas[1,7]

[1]Institute of Molecular Biology and Biotechnology, Foundation for Research and Technology—Hellas, Heraklion 70013, Greece
[2]Department of Biology, University of Crete, Vassilika Vouton, Heraklion 70013, Greece
[3]Laboratory of Biological Chemistry, School of Medicine, Faculty of Health Sciences, Aristotle University of Thessaloniki, Thessaloniki, Greece
[4]Functional Proteomics and Systems Biology (FunPATh), Center for Interdisciplinary Research and Innovation (CIRI-AUTH), Balkan Center, Thessaloniki, Greece
[5]Instituto de Investigaciones Bioquímicas de La Plata, Centro Científico Tecnológico La Plata, Consejo Nacional de Investigaciones Científicas y Técnicas—Facultad de Ciencias Médicas, Universidad Nacional de La Plata, La Plata 1900, Argentina
[6]Department of Vector Biology, Liverpool School of Tropical Medicine, Liverpool L3 5QA, UK
[7]Pesticide Science Laboratory, Department of Crop Science, Agricultural University of Athens, 11855 Athens, Greece

VB, 0000-0001-7337-4765; JV, 0000-0002-8704-2574

Malaria incidence has halved since the year 2000, with 80% of the reduction attributable to the use of insecticides. However, insecticide resistance is now widespread, is rapidly increasing in spectrum and intensity across Africa, and may be contributing to the increase of malaria incidence in 2018. The role of detoxification enzymes and target site mutations has been documented in the major malaria vector *Anopheles gambiae*; however, the emergence of striking resistant phenotypes suggests the occurrence of additional mechanisms. By comparing legs, the most relevant insect tissue for insecticide uptake, we show that resistant mosquitoes largely remodel their leg cuticles via enhanced deposition of cuticular proteins and chitin, corroborating a leg-thickening phenotype. Moreover, we show that resistant female mosquitoes seal their leg cuticles with higher total and different relative amounts of cuticular hydrocarbons, compared with susceptible ones. The structural and functional alterations in *Anopheles* female mosquito legs are associated with a reduced uptake of insecticides, substantially contributing to the resistance phenotype.

## 1. Introduction

Malaria is a life-threatening disease causing more than 500 000 deaths annually in sub-Saharan Africa, mostly in children under five and pregnant women. Prevention of the disease is best achieved by vector control which, nowadays in Africa, largely relies on contact insecticides. However, the evolution and spread of insecticide resistance seriously threatens the success and sustainability of control interventions, and it is possibly associated with the recent rise of malaria cases in 2018, for the first time since 2000.

Overexpression of detoxification enzymes, which inactivate or sequester insecticides, and mutations in the target site that alter the affinity of insecticide binding have been widely described in malaria vectors, however, may not explain the recent emergence of striking multiple-resistant phenotypes in West Africa [1].

Proc. R. Soc. B 286: 20191091

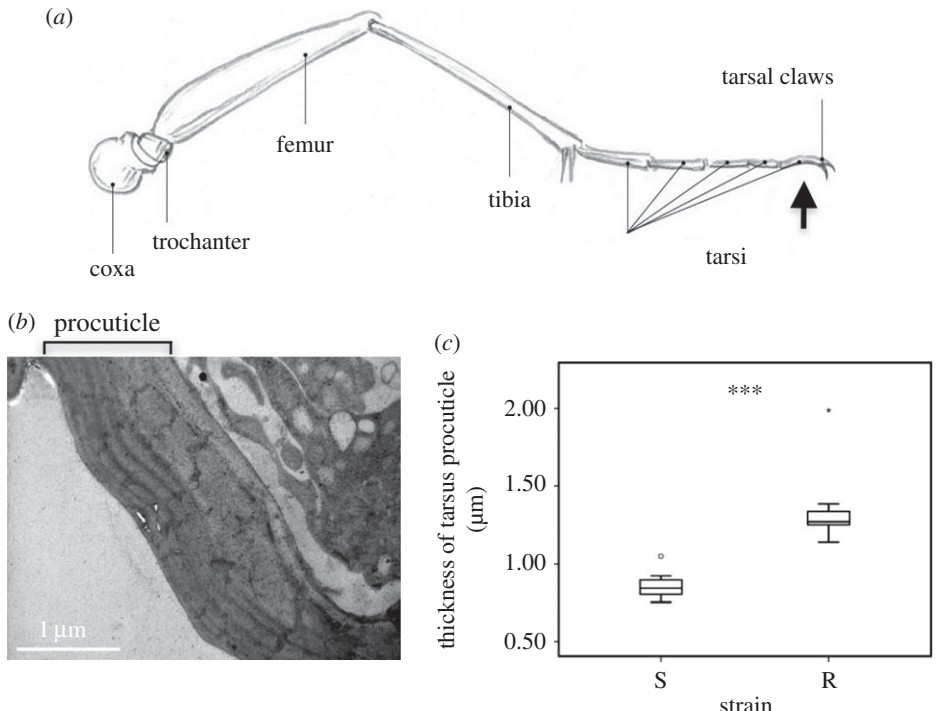

**Figure 1.** TEM analysis of tarsus thickness. (*a*) Image of mosquito leg parts adapted from https://johnmuirlaws.com. The black arrow indicates in which leg part sections were taken. (*b*) A representative TEM image from a cross-section of the tarsus cuticle. (*c*) Box-and-whisker plot of cuticle thicknesses. The boxes represent the 25% and the 75% percentiles of five independent measurements for resistant (R, right box) and susceptible (S, left box) legs. The horizontal black line within each box indicates the median; error bars correspond to the 10th and 90th percentiles. ***$p$-value $< 0.001$ determined by ANOVA.

Insecticides must first bypass both cellular and non-cellular leg barriers to reach their targets within the neuronal cells, the voltage-gated sodium channels [2,3]. Mosquito legs play a major physiological role in their life. They have been adapted to implement diverse biological processes, ranging from walking on top of water or downwards on ceilings to sensing mechanical or chemical stimuli [4]. The outermost part of the mosquito leg is the cuticle serving a variety of functions like protection from desiccation, chemical communication and sensory perception of the environment, mechanical support and locomotion. The leg cuticle is also the first barrier protecting the insect from the penetration of external compounds, such as insecticides [5].

The cuticle is sub-divided into two layers: the 'inner' procuticle, which forms the bulk of the cuticle, and the outermost layer, called the epicuticle. The procuticle contains chitin filaments complexed with cuticular proteins (CPs) [6,7], but how exactly these proteins interact to each other or with chitin remains elusive. A plethora of CPs belonging to diverse protein families (CPR, CPAPn, CPG, CPF, CPLCG and more) have been identified in many insects so far, including mosquitoes [8]. The majority of CPs belong to the CPR family, with the characteristic Rebers and Riddiford (R&R) Consensus that confers extended chitin-binding properties [7]. Recent studies show that CP members contribute to cuticle maintenance and structure, since silencing of specific genes resulted in thinner and/or malformed cuticles [9,10].

The epicuticle surface is covered by lipids, the majority of which are hydrocarbons (CHCs), free fatty acids and wax esters, while highly tanned proteins associated with lipids are located in inner layers [11–13]. CHCs are the most abundant lipid species in the *Anopheles* epicuticle, forming a complex mixture of multi-isomeric methyl- and dimethyl-branched components, as well as straight-chain alkanes or monounsaturated alkenes [14]. The final step of CHC production is catalysed by the insect-specific CYP4Gs in specialized cells, the oenocytes, located in abdominal walls [15,16]. Upon biosynthesis, large amounts of CHCs are bound to lipophorins in the haemolymph and are finally transported to different body tissues, such as oocysts, as well as being transferred for deposition on insect surface [17–21]. The exact deposition mechanism, including uptake and travelling through epidermal cells and integument, remains unknown. On top of the cuticle, CHCs form a highly water-proofing layer, which confers desiccation resistance [22–24]. Moreover, different CHC blends are characteristic for different insect species, whereas the species-specific CHC signature may vary quantitatively because of different geographical origins [25]; they can also discriminate age and/or sex within species [26–30], as well as playing dual roles in desiccation resistance and pheromonal communication [31].

A role of cuticle thickening in conferring insecticide resistance, one of the most urgent WHO concerns for malaria control, has previously been demonstrated. A strain of *An. gambiae* originating from West Africa, resistant to pyrethroids and DTT, had thicker procuticle in the femur leg segment, a phenotype correlated to over-expression of CPLCG3 and CPRs [32]. In *Culex pipiens pallens*, the femur cuticle was also thicker in resistant versus susceptible populations, mainly because of CPLCG5, since gene-specific silencing resulted in a thinner cuticle and increased susceptibility upon insecticide contact [10,33]. Additionally, a multi-resistant *An. gambiae* mosquito strain from Côte d'Ivoire (Tiassale) had a thicker femur cuticle (figure 1*a*), primarily due to enriched deposition of CHCs, which delayed the penetration rate of contact insecticides, and thus presumably produced a more intense and broader insecticide resistance phenotype [14,34]. Noteworthily, CHCs were enriched in Tiassale-resistant *An. gambiae*, compared to a susceptible population, at the whole-body level, a difference

attributed primarily to quantitative, but not qualitative changes [14]. All the above cases report cuticular thickness changes in femurs, although these leg compartments do not come in direct contact with insecticide molecules *per se*. Insecticide contact would be mediated predominately by the tarsi, and hence these most distal leg segments are expected to play the most important role in insecticide uptake (figure 1*a*). However, changes in this leg compartment have not been studied to date.

So far, cuticle alterations have been associated with insecticide resistance in *An. gambiae* and other insects, but the cellular and molecular underpinnings of this phenomenon remain largely unclear. Transcriptomic analysis indicated the association of elevated expression of both cuticle genes and genes associated with the hydrocarbon biosynthetic pathway (such as CYP4Gs, elongases, fatty acid synthetases), with the resistance phenotype in *An. gambiae* and *An. arabiensis* [35,36], while the role of CYP4G16 in the biosynthesis of *An. gambiae* CHCs was functionally confirmed [14]. However, the detailed mechanism responsible for lipid transport and epicuticular deposition remains unknown.

Here, by using transmission electron microscopy (TEM), and lipidomic and proteomic approaches, we have studied structural and functional alterations in the legs of highly resistant *An. gambiae*, compared to susceptible counterparts.

## 2. Material and methods

### (a) Mosquito strains

The *An. gambiae* N'Gousso (NG) strain from Cameroon is susceptible to all classes of insecticides, whereas the *An. gambiae* VK7 strain from Burkina Faso (VK7) is resistant mainly to pyrethroids and DDT [37]. Mosquitoes were reared under standard insectary conditions at 27°C and 70–80% humidity under a 12 h : 12 h photoperiod. Both strains have been maintained overtime under the same laboratory conditions before analysis.

### (b) Transmission electron microscopy

The cuticle thickness of mosquito legs was measured by TEM, as previously described [14]. Before TEM, one wing from every individual was removed, measured and used as proxy for body size [38]. Individuals with similar wing size were selected and further analysed by TEM. Ultra-thin gold sections of the tarsi, the most distal leg segments immediately after the two claws, were taken from groups of 25 female mosquitoes (resistant and susceptible), 3–5 days old, and observed under a high-resolution JEM 2100 transmission electron microscope (JEOL), at an operating voltage of 80 kV.

Raw images from electron microscope were analysed in IMAGE J (v. 1.52e) and the results from at least five independent measurements are presented as means ± standard error (±s.e.). The statistical analysis of cuticle thickness of both resistant and susceptible legs was performed by SPSS software tool v. 22 for Windows (SPSS Inc., Chicago, IL, USA). Differences in the cuticle thickness between the two samples (resistant and susceptible) were regarded significant (ANOVA, $p < 0,001$).

### (c) Extraction of lipids and cuticular hydrocarbon fractionation, identification and quantitation

Cuticular lipids from 5-day-old-females and males both from resistant and susceptible populations (non-blood fed)—female resistant (FR), male resistant (MR), female susceptible (FS) and male susceptible (MS)—were extracted by 1-min immersion in hexane (x3) with gentle agitation; extracts were pooled and evaporated under an $N_2$ stream (10 insects per tube, $n = 7$ for each group of mosquitoes). CHCs were separated from other components by adsorption chromatography on a minicolumn (2.5 × 0.5 cm i.d.) of activated SUPELCOSIL A (Supelco), eluted with hexane (4 ml) and then concentrated under an $N_2$ atmosphere. A similar procedure was used for the extraction and fractionation of leg CHCs (60 legs per tube, $n = 7$ for each group of mosquitoes). All seven biological replicates were collected from three serial mosquito generations. CHC identification by gas chromatography-mass spectrometry (GC-MS) and CHC quantitation by GC-flame ionization detector (FID) were performed as described previously [14,39].

Statistics were analysed using GraphPad PRISM software, v. 5.03. Differences in the total CHC values were analysed with both Student's *t*-test and one-way ANOVA.

### (d) Classification and principal component analysis analysis of cuticular hydrocarbons

Relative abundance of 70 CHC species was measured for 14 female mosquito samples (pools of 10 individuals per sample), and finally 35 CHC species (out of 70) that satisfied the abundance criteria (greater than 0.1%) were selected. Seven samples originated from the VK7 (resistant) mosquito population and the remaining seven from the NG (susceptible) mosquito strain. CHCs were isolated from two different body parts from each mosquito pool: the legs and the rest of the body (excluding legs). In all, our dataset consisted of 35 CHC measurements assigned to (a) body part (legs or bodies) and (b) susceptibility (susceptible or resistant).

Abundances were normalized through quantile normalization and were initially subjected to an exploratory PCA analysis using different combinations of target variables (including susceptibility and body parts), which suggested the existence of particular CHCs that could be used as discriminatory features for body part of origin and susceptibility in female samples. Subsequently, a random forest classification strategy was employed (electronic supplementary material) and the abundances of resulting CHCs were subjected to the final PCA analysis.

### (e) Comparative proteomics analysis by nLC-MS/MS

Leg protein extracts from both mosquito strains were prepared using the solubilization buffer (SB: 8 M Urea, 0.2 M NaCl, 0.1% SDS, 50 mM Tris–HCL, pH 8.0). Briefly, legs from 25, 3–5-day-old-female and non-blood fed mosquitoes, were dissected and mechanically homogenized in SB by hand pestles. Upon centrifugation (2500 g for 15 min at 10°C), the pellet was discarded and the protein samples were separated on 10% polyacrylamide gels, following staining with 0.1% Coomassie Brilliant blue R-250 (in 40% methanol 10% acetic acid) and destaining with 10% ethanol, 7.5% acetic acid [40]. Each lane was excised and cut in 15 bands, which were subsequently destained, reduced, alkylated and in-gel tryptic digested as previously described [41]. Tryptic peptides were finally dried in a speed-vacuum centrifuge and dissolved in 5% FA in ultra-pure water solution. All samples were desalted by being loaded on conditioned home-made pre-columns packed with C18 extraction disks (Empore) and eluted stepwise with 80% MeOH and 5% FA. All elution fractions were collected, speed-vacuum centrifuged and diluted in 5% FA for further nLC-MS/MS analysis.

Protein identification and relative quantitation by nLC-MS/MS was done on a LTQ-Orbitrap XL coupled to an Easy nLC (Thermo Scientific). The sample preparation and the LC separation were performed as previously described with minor modifications [42]. Briefly, the dried peptides were dissolved in

20 µl 0.5% formic acid, aqueous solution and the tryptic peptide mixtures were separated on a reversed-phase column (ReprosilPur C18 AQ, particle size = 3 µm, pore size = 120 Å, Dr Maisch), fused silica emitters 100 mm long with a 75 µm internal diameter (Thermo Scientific), packed in-house, using a pressurized (35–40 bars of helium) packing bomb (Loader kit SP035, Proxeon). The nLC flow rate was 200 nl min$^{-1}$. Tryptic peptides were separated and eluted in a linear water-acetonitrile gradient and injected into the mass spectrometer [42,43]. MS survey scans were acquired in the Orbitrap from 200 to 2000 $m/z$ at a resolution of 60 000 and for the MS/MS, precursor isolation at 1.6 $m/z$ was performed by the quadrupole (Q). Fragmentation of 20 most intense ions by collision-induced dissociation (CID) with normalized collision energy of 35% and rapid scan MS analysis were carried out in the ion trap. The dynamic exclusion duration was set to 15 s with 10 ppm tolerance around the selected precursor and its isotopes. The AGC target values were set to $4.0 \times 105$ and $1.0 \times 104$ and maximum injection times were 50 ms and 35 ms for MS and MSn scans, respectively.

MS raw data were analysed by Proteome Discoverer 1.4.0 (Thermo Scientific) using MASCOT 2.3.01 (Matrix Science) search algorithm. Spectra were run against the latest version of *An. gambiae* theoretical proteome containing 13 515 entries (UniProt 2018) and a list of common contaminants [44,45]. Search parameters employed are described in detail elsewhere [46]. Final peptide and protein lists were compiled in SCAFFOLD (v. 4.4.1.1, Proteome Software; Portland, OR, USA) employing criteria previously described [46]. Manual annotation of identified proteins was also performed.

To determine the differentially expressed peptides, we compared the abundances of the peptides between resistant and susceptible control mosquito legs, using a minimum of 3.0-fold changes (normalized to median). GO terms and pathways enrichment analysis of the differentially expressed proteins, based on fold change criteria, was performed using Gene Ontology Consortium [47]. In addition, a beta-binomial test, using the ibb library in R, was employed to identify significant changes at single-protein level between the control and resistant samples among the three biological replicates. $p$-values < 0.1 were considered to be significant.

## (f) Western blot analysis

Legs from twenty-five, 3–5-day-old female mosquitoes were dissected and mechanically homogenized in solubilization buffer (SB: 8 M urea, 0.2 M NaCl, 0.1% SDS, 50 mM Tris–HCL, pH 8.0). Upon centrifugation (2500 g for 15 min at 10°C), pellet was discarded and the supernatant was analysed by SDS-PAGE and western blot. Antibody against b-Tubulin (E-10, 1 : 500) was from Santa Cruz Biotechnology (SC-365791), and antibody against CPCFC1 (1 : 500) was the anti-AgamCPCFC1 produced in Vannini *et al.* 2017 [7] and were kindly provided by J. Willis.

## (g) Quantitation of chitin

Chitin determination in mosquito legs was made according to Lehmann & White [48] and subsequent modifications [49–51]. Briefly, legs and remaining bodies from both mosquito strains (susceptible and resistant) in three replicates of eight female mosquitoes, each 3–5 days old (48 legs each replicate), were mechanically homogenized and further processed for deacetylation of chitin to chitosan and determination of chitosan (i.e. glucosamine polymer). Finally, samples were transferred to a 96-well microplate and absorbance was determined at 650 nm in a plate reader (Molecular Devices, Spectra Max). Before chitin quantification, mosquitoes' leg and body weights were measured and used for normalization.

# 3. Results

## (a) Pro-cuticle is thicker in the leg tarsus of resistant *An. gambiae* females

TEM analysis of cuticle thickness in resistant (VK7) and susceptible (NG) female mosquito legs focused in the leg tarsi, the most physiologically relevant segment of the leg regarding insecticide uptake, since mosquitoes contact insecticide treated surfaces with their tarsi. Ultra-thin cross-sections from leg tarsi right after the two claws of both strains were taken and measurements in each section (greater than 10) were done randomly and blinded to TEM operator. The overall pro-cuticle thickness was found to be significantly higher in the resistant strain (1.3237 µm ± 0.05) when compared with the susceptible one (0.8652 µm ± 0.03; $p < 0.001$; figure 1c). Under the experimental conditions, the epicuticle could not be stabilized in the tarsi and therefore was not measured directly, but determined indirectly via the CHC analysis.

## (b) Total cuticular hydrocarbons and relative amounts of specific cuticular hydrocarbons are altered in insecticide-resistant *An. gambiae* female mosquitoes

### (i) Cuticular hydrocarbons are specifically enriched in the legs of resistant *An. gambiae* female mosquitoes

Resistant female legs had 43.15% more CHCs (the most abundant lipid species in the *Anopheles* epicuticle) compared to the control legs (mean CHC levels: 286.77 ng mg$^{-1}$ ± 48.74 and 163.03 ng mg$^{-1}$ ± 15.70, respectively; figure 2a). The levels of CHCs from the legs of male resistant mosquitoes were 32.31% higher compared to the susceptible ones (mean CHC levels: 224.46 ng mg$^{-1}$ ± 17.01 and 151.93 ng mg$^{-1}$ ± 13.44, respectively; figure 2a). On whole organism extracts, resistant female mosquitoes have 16.37% more total CHCs than the susceptible control mosquitoes (mean CHC levels: 964.98 ng mg$^{-1}$ ± 134.42 and 806.97 ng mg$^{-1}$ ± 53.12, respectively; figure 2b), and this difference is similar in the male mosquitoes, 16.67% (mean CHC levels of resistant males: 965.91 ng mg$^{-1}$ ± 28.43 and mean of susceptible males: 804.9 ng mg$^{-1}$ ± 57.41).

Interestingly, 29.66% of the total CHCs present in resistant *An. gambiae* whole female insects are found in legs, whereas a lower proportion of total CHC content (20.17%) was detected in the legs of susceptible females. In addition, resistant and susceptible male mosquitoes had 23.28% and 18.98% of total CHCs in their legs, respectively. These data suggest that resistant mosquitoes have more abundant hydrocarbons deposited in their cuticle and that this is especially true in the female leg.

### (ii) Legs are differentiated from the rest of the body based on the relative amounts of selected cuticular hydrocarbon components

To further understand the CHC features that are present in the legs, a random forest approach (RF) was conducted and followed by principal component analysis (PCA). Legs were dissected from the rest of the bodies and 70 CHC species were relatively quantified in total (legs and bodies from 10 female mosquitoes in each replicate were analysed, seven replicates totally), from both resistant and susceptible mosquitoes. Thirty-five CHCs (single or multicomponent peaks) with

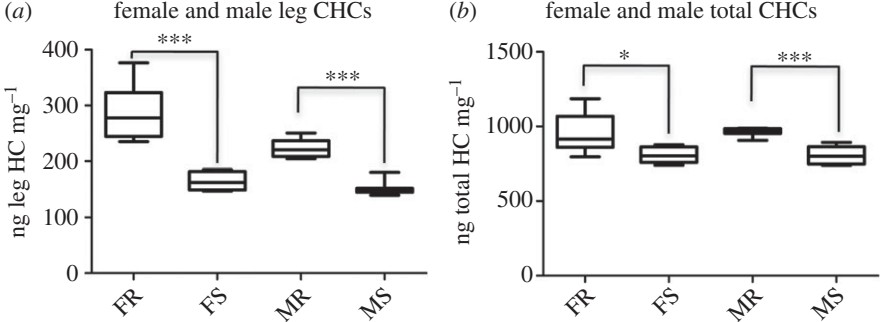

**Figure 2.** Quantitation of CHCs in *An. gambiae* mosquitoes. (*a*) Quantitation analysis of CHCs in mosquito legs of resistant female and male mosquitoes, compared to susceptible ones. ***$p < 0.001$ in both cases as determined by *t*-test. (*b*) Quantitation analysis of CHCs in whole mosquitoes, as in (*a*). *$p < 0.05$ for females and ***$p < 0.001$ for males, determined by *t*-test. Seven biological replicates were analysed in both (*a*) and (*b*). Data are presented as box-and-whisker plots. The boxes represent the 25% and 75% percentile and the black lines within the boxes indicate the medians; error bars correspond to the 10th and 90th percentiles. FR, female resistant mosquitoes; FS, female susceptible mosquitoes; MR, male resistant mosquitoes; MS, male susceptible mosquitoes.

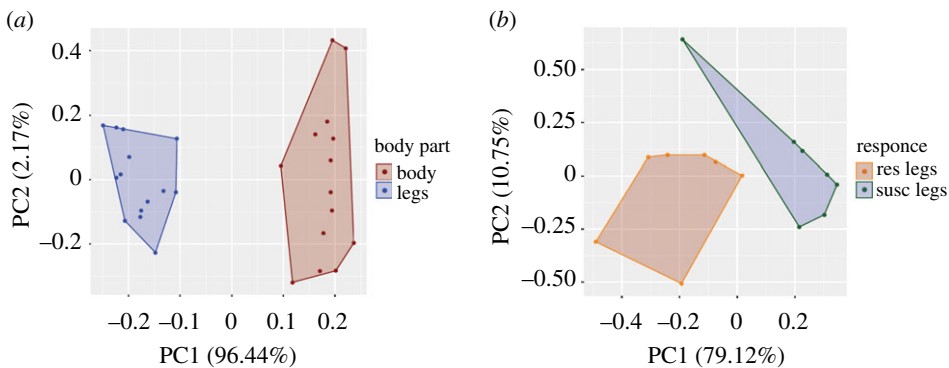

**Figure 3.** Classification and PCA analysis of CHCs in legs and bodies of resistant and susceptible female *An. gambiae* mosquitoes. (*a*) PCA plots of CHCs in bodies and legs from female mosquitoes, both resistant and susceptible (shown in different colours), using the top predictors above the threshold in electronic supplementary material, figure S1A. (*b*) PCA visualization of classification of CHCs in legs from susceptible (S) versus resistant (R) mosquitoes, using the top predictors above the threshold in electronic supplementary material, figure S1B. (Online version in colour.)

relative abundance greater than 0.1% were selected for the analysis (RF and PCA). According to the best out of 1000 RF models, six multicomponent CHC peaks were selected as best predictors and further used for classification visualized in PCA plots (electronic supplementary material, figure S3A). Five of them contain a number of dimethyl branched isomers, coeluting as dimethyl C40, dimethyl C39, dimethyl C37, dimethyl C43 and dimethyl C45, and one is a mono-methyl branched multicomponent peak (methyl C41; electronic supplementary material, table S1). Furthermore, by using the above six CHC peaks, the first two PCs were sufficient to discriminate body parts (legs and bodies without legs) almost perfectly (98.61%; figure 3*a*). In addition, heat map and hierarchical clustering confirmed discrimination based on the relative abundances of the particular CHCs (electronic supplementary material, figure S4A).

### (iii) Different relative amounts of cuticular hydrocarbon species found in resistant mosquito legs compared to susceptible

The next question to be answered was whether resistant female legs differed from the susceptible control legs. Using the same RF approach as above (see Material and methods for details), six CHC peaks were defined as the most important for classification of leg samples into resistant or susceptible (electronic supplementary material, figure S3B). Three CHCs are straight-chain alkanes, n-C30, n-C28 and n-C29, whereas the other CHCs correspond to a monounsaturated

straight-chain alkene (n-C31:1) coeluting with a mono-methyl branched alkane (3-methyl C30) and two di-methyl branched alkanes (dimethyl C39 and dimethyl C41; electronic supplementary material, table S2). Finally, PCA classification using the six CHCs showed a clear discrimination between resistant and susceptible female legs (first two PCs amounting to 89.87% of the variance; figure 3*b*), confirmed also by heat map and hierarchical clustering (electronic supplementary material, figure S4B).

### (c) Proteomic analysis and biochemical validation

In order to understand underlying biochemical changes responsible for the enriched leg cloak in the resistant female mosquitoes and other changes possibly associated with resistance in the legs of resistant mosquitoes, the leg proteome, consisting of 1120 proteins, was characterized (electronic supplementary material, figure S1). Quantitative differences in the leg proteome of the resistant and susceptible female mosquitoes compared to susceptible ones were determined by label-free quantitative proteomics (electronic supplementary material, figure S1).

### (i) Cuticle proteins and elevated chitin content associated with the thicker procuticle in resistant mosquitoes

Structural constituent of cuticle was the most enriched cluster in the pyrethroid resistant leg proteome, compared to susceptible. The structural CPs showed a substantial 5.1 enrichment ratio in the set of the over-expressed proteins. From the 41

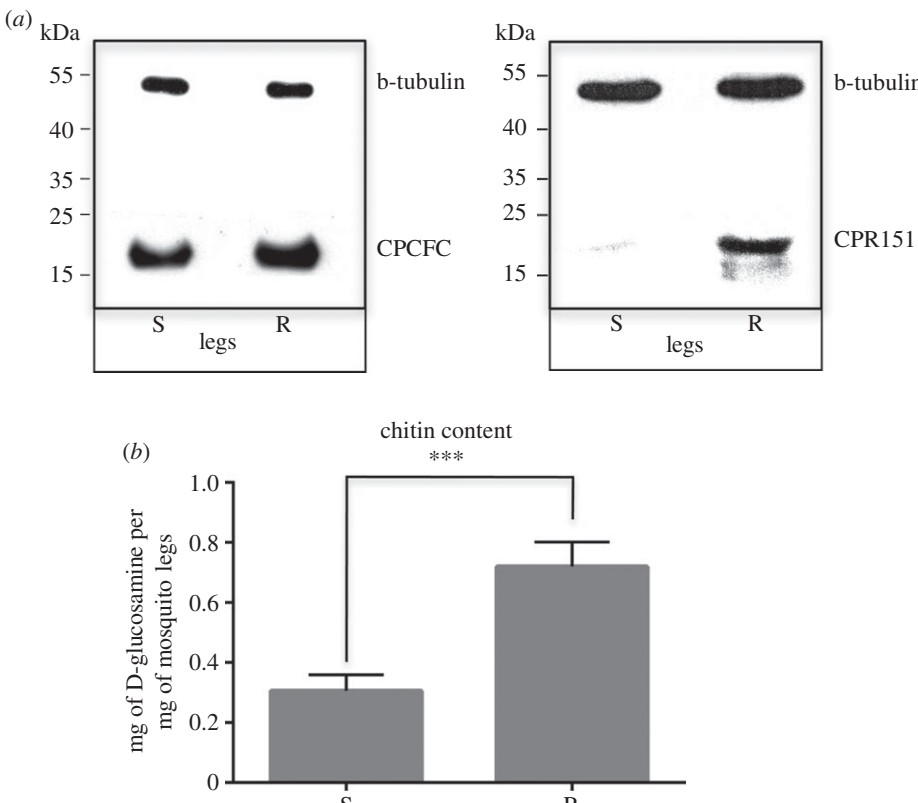

**Figure 4.** Cuticular proteins and chitin content in legs of resistant *An. gambiae* female mosquitoes. (*a*) Western blots to confirm differential protein expression levels (anti-CPFC1, left blot and anti-CPR151, right blot). Anti-b-tubulin was used in the same blots for loading control. S, susceptible and R, resistant. (*b*) Chitin quantitation (mg of D-glucosamine per mg of mosquito legs) in resistant (R: VK7) compared to susceptible legs (S: NG). The legs from eight female mosquitoes were analysed in both cases and measurements were repeated three times (48 legs per replicate). Significance determined by *t*-test (two-tailed *p*-value < 0.001, ***) and the data are presented as means + s.e.m.

CPs detected in the leg proteome, 31 were differentially regulated (29 proteins were over-expressed and only CPR8 and CPR120 were decreased; electronic supplementary material, figure S2A,B). Furthermore, over 65% of the differentially expressed CPs belonged to the CPR family, with characteristic chitin-binding motifs (electronic supplementary material, figure S2B). The protein expression levels of the CPR151 and the CPCFC1 CPs was also analysed by western blot analysis, which confirmed their over-expression in the protein extracts of resistant legs (figure 4*a*).

To determine whether these proteome differences were also reflected in altered leg physiology, we quantified chitin in the legs, according to the protocol developed by Lehmann *et al.* [48]. Indeed, a substantially larger amount of chitin monomer D-glucosamine was found to be present in the leg cuticle of insecticide-resistant female mosquitoes (0.71 mg of D-glucosamine per mg of mosquito legs ± 0.03) compared to the susceptible mosquitoes (0.3 mg of D-glucosamine per mg of mosquito legs ± 0.02; *p* < 0.0001; figure 4*b*). This difference was diminished when D-glucosamine from body extracts was measured in both resistant (1.46 mg of D-glucosamine per mg of mosquito bodies ± 0.49) and susceptible mosquitoes (1.12 mg of D-glucosamine per mg of mosquito bodies ± 0.16; electronic supplementary material, figure S5). Chitin synthase 1 protein that catalyses the last step in the production of chitin was not identified in the leg proteome. However, quantitative RT-PCR detected significant upregulation of chitin synthase 1 (chs1) mRNA levels by 3.86-fold in the resistant compared to the control legs (electronic supplementary material, figure S6). Chitin synthase 2 is expressed at very low levels in mosquito legs and differences in expression were not examined.

## 4. Discussion

This study describes substantial differences in the properties of the insect cuticle in two strains of *An. gambiae* that differ in their insecticide resistance profile. Although these strains are from geographically distinct locations and have been maintained in colony for multiple years, the compelling evidence for quantitative differences in the cuticle both in the total amounts as well as in the relative amounts of certain components of the resistant strain used in this study is supportive of previous work demonstrating the importance of reduced penetration through the cuticle as a key resistance mechanism in *An. gambiae*. Importantly, this study highlights the critical role of cuticle composition in legs, as the body part that contacts insecticide used in indoor applications for malaria control, and provides novel insights into the putative mechanisms of cuticular resistance.

We have demonstrated that changes in CHC content, as well as structural procuticle components, could be responsible for building a thicker leg cloak in the tarsus of VK7, a highly resistant *An. gambiae* population from W. Africa. The legs from resistant mosquitoes, in particular, had a substantially higher CHC content, compared to susceptible mosquitoes. A difference in the content of CHCs had been previously reported at the whole body level of another resistant population (Tiassale) from Côte D' Ivoire [14], but the current study shows that this difference is indeed primarily attributed to changes in the legs, the most physiologically relevant body part. Moreover, the accumulation of CHC mass in legs of resistant mosquitoes possibly implicates transport mechanisms towards the legs and/or towards the leg's epicuticle, since CHC biosynthesis

probably occurs in the abdominal oenocytes [16]. Thus, local concentration of specific CHC binding molecules like lipophorins and pheromone-binding proteins (PBPs) in the leg's haemolymph and the expression of their specialized receptors is increased and/or transporter molecules such as ABC transporters expressed in the leg's epidermis are more abundant or active. Additionally, we found six methyl-branched CHC peaks as the most differentiated CHCs, in terms of relative amounts, compared to the CHC reservoir of the remaining body, also implicating a selective transport of CHCs to the legs. Selective transport pathways for the export of various CHC components to the epicuticle and to different tissues, have been suggested in many studies [17,52], including specific transport of volatile sex-pheromone in moths and social ants [53,54].

Along with the overall quantitative differences, resistant legs showed a characteristic CHC signature based on their relative concentrations, when compared to the susceptible control legs. Six CHC species were particularly over-represented in the resistant legs compared to susceptible ones. Moreover, CHCs as major components of the insect epicuticle exert a broad spectrum of functions affecting the whole insect's physiology. Generally, very long-chain alkanes contribute to the waterproofing properties of the cuticle lipid layer and, ultimately, to desiccation tolerance [55]. Very long-chain alkenes and methyl-branched alkanes might increase the chemical information content of the cuticle, since their role in chemical communication in various insect species is well documented (for example 7, 11-heptacosadiene is the main excitatory pheromone in *Drosophila melanogaster* females). CHCs have been shown to play dual roles in *Drosophila*, both in pheromonal communication and in tolerance to desiccation [56,57]. Thus, the mating capacity/fitness of resistant mosquitoes might be affected by the over- or down-representation of specific leg CHCs, although this possibility needs to be further investigated.

Subsequently, the leg proteome was analysed by gel-based proteomics. It consisted of 1120 proteins, with energy production and lipid metabolism proteins enriched compared to the *Anopheles* whole body proteome. Comparative proteomic analysis between legs from insecticide resistant and susceptible *An. gambiae* mosquitoes showed that the most upregulated proteins were the structural CPs, in agreement with a leg-thickening phenotype. The over-expression of CPs in the cuticle of resistant insects has been also reported in other insects/mosquitoes at the whole organism level and it has been associated with reduced insecticide penetration and thicker cuticles [58,59]. In addition to the large number of substantially elevated CPs, resistant mosquitoes also had substantially elevated chitin levels in their legs (and not in their remaining bodies) which apparently bound to the elevated CPs and reinforced procuticle chitin-rich layers in the legs. Furthermore, upregulation of chs1 transcripts could be correlated with increased production of chitin in legs of resistant compared to the susceptible female mosquitoes. Noteworthily, two CPs, CPR8 and CPR120, were found to be downregulated in the resistant legs, indicating that not only thickening but also qualitative differences could affect insecticide penetration by a yet unidentified mechanism.

The presence and/or differential expression of protein families which have been previously associated with insecticide resistance were also identified (electronic supplementary material, table S3), such as ABC transporters, detoxification enzymes and odorant binding proteins (OBPs). ABC transporters of subfamily G have been implicated in lipid transport to the cuticle epidermis [60]; however, we have identified AGAP003680-PA expressed in leg proteome, an ABC transporter that belongs to subfamily H. This transporter has been found differentially expressed in a number of insecticide-resistant populations of *An. gambiae* across Africa [61]. The role of ABCHs in insects remains largely elusive; however, silencing of one of them in *Tribolium castaneum* (*Tc*) adults (*Tc*ABC-9C) resulted in a reduction of cuticle lipids, suggesting a possible role in lipid transport towards the cuticle [62]. Moreover, the Snu and ABCH-9C transporters from *Drosophila melanogaster* and *Locusta migratoria* (*Lm*) were shown also to participate in the construction of lipid-based barrier of cuticle [63]. Phylogenetic analysis classified *Lm*ABCH-9C, *Tc*ABC-9C and AGAP003680 in the same group (group C) of ABCH subfamily [64].

A number of detoxification enzymes, such as P450s, hydrolases, carboxyl esterases, glutathione-s-transferases and antioxidant enzymes, were also expressed in mosquito legs. Among them, superoxide dismutase (CuSOD3) and glutathione-s-transferases (GSTe3 and GSTmic1), found significantly upregulated in the resistant legs, are well-known antioxidant enzymes that protect living cells from oxidative damage, including pyrethroid-induced lipid peroxidation [65–67]. Interestingly, seven OBPs, which can bind small organic compounds and possibly act as scavengers for insecticides [68–70], were differentially regulated in the legs of resistant *An. gambiae* mosquitoes. We hypothesize that the upregulation of two OBPs in the proteomic dataset (i.e. OBP10: 6.33-fold and OBP57: 3.67-fold) might be associated with the increased binding and transport of the extra CHCs found in the resistant legs [71], and/or sequestration of insecticides, although further studies are required to support this hypothesis. Conversely, the downregulation of five OBPs (OBP13, OBP5470 and three putative D7-related proteins) could possibly indicate an altered chemoreception mechanism in the legs of resistant *An. gambiae*, for example, via a reduction of hydrophobic channels facilitating the transport of ligands to their receptors, as previously suggested [72,73].

In conclusion, *An. gambiae* female mosquitoes seem to build a protective leg cloak to fight against insecticides. Reconstruction of thicker and/or altered leg cuticle, via extra deposition of chitin, proteins and CHCs, act in concert to fight, in first line, against insecticides. These changes might be associated with other major physiological functions in *An. gambiae* life history, such as pheromonal communication and resistance to desiccation, thus affecting fitness and positive selection of insecticide resistance in the field.

Data accessibility. All data generated or analysed during this study are available in this manuscript (and its electronic supplementary material).

Authors' contributions. V.B. and J.V. conceived and designed the experiments; V.B., M.K., V.K., J.R.G., S.J.M., E.P. and A.K performed the experiments; V.B., M.K. and J.R.G. carried out the statistical analysis and C.N. performed the computational analysis; M.A. and S.J.M. performed the mass spectrometry analysis. V.B., M.P.J., G.C., H.R. and J.V. wrote the manuscript.

Competing interests. We declare we have no competing interests.

Funding. This project has received funding from the Hellenic Foundation for Research and Innovation (HFRI) and the General Secretariat for Research and Technology (GSRT), under grant agreement no. 2040 (V.B. and V.K.). Also this study is co-financed by Greece and the European Union (European Social Fund) through the operational programme 'Human Resources Development, Education and Lifelong Learning' in the context of the project 'Strengthening Human Resources Research Potential via Doctorate Research' (MIS-5000432), implemented by the State Scholarships Foundation (IKΥ) (M.K.), by the Hellenic Foundation for Research and Innovation (HFRI) and the General Secretariat for Research

and Technology (GSRT), under the HFRI PhD Fellowship grant (grant agreement no. 91.0005) (N.K.) and the European Union's Horizon (INFRAVEC) research and innovation programme under grant agreement no. 731060 (J.V.). Support is also acknowledged to the Argentine National Scientific and Technological Research Council (CONICET) (PIP 150100390) (M.P.J.).

Acknowledgements. We thank Prof. Judith Willis for providing the antibodies against CPs.

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
