## [Reviewer comments · Proceedings of the Royal Society B: Biological Sciences]

Review History

RSPB-2019-0558.R0 (Original submission)

Review form: Reviewer 1

Recommendation

Accept as is

Scientific importance: Is the manuscript an original and important contribution to its field?

Excellent

General interest: Is the paper of sufficient general interest?

Excellent

Quality of the paper: Is the overall quality of the paper suitable?

Excellent

Is the length of the paper justified?

Yes

Should the paper be seen by a specialist statistical reviewer?

No

Do you have any concerns about statistical analyses in this paper? If so, please specify them explicitly in your report.

No

It is a condition of publication that authors make their supporting data, code and materials available - either as supplementary material or hosted in an external repository. Please rate, if applicable, the supporting data on the following criteria.

Is it accessible?

N/A

Is it clear?

N/A

Is it adequate?

N/A

Do you have any ethical concerns with this paper?

No

Comments to the Author

The authors are to be congratulated for a highly significant innovative study that shows how insecticide resistant mosquitoes structurally and physiologically build a thickened leg cloak in response to insecticide exposure. Research to identify mechanisms is excellent and carefully done. This is a landmark paper that opens a new field of research. No suggestions for improvement, except very minor edits.

Review form: Reviewer 2

Recommendation

Reject – article is not of sufficient interest (we will consider a transfer to another journal)

Scientific importance: Is the manuscript an original and important contribution to its field?

Marginal

General interest: Is the paper of sufficient general interest?

Marginal

Quality of the paper: Is the overall quality of the paper suitable?

Acceptable

Is the length of the paper justified?

Yes

Should the paper be seen by a specialist statistical reviewer?

No

Do you have any concerns about statistical analyses in this paper? If so, please specify them explicitly in your report.

No

It is a condition of publication that authors make their supporting data, code and materials available - either as supplementary material or hosted in an external repository. Please rate, if applicable, the supporting data on the following criteria.

Is it accessible?

Yes

Is it clear?

Yes

Is it adequate?

Yes

Do you have any ethical concerns with this paper?

No

Comments to the Author

In their work "Mosquito build leg cloak to fight against insecticides", Vasileia Balabanidou and colleagues report on the resistance of *Anopheles gambiae* to insecticides through changes in the tarsal cuticle thickness and composition.

Overall, reading of the manuscript was cumbersome; this may be a matter of style though. Before commenting the data of this work, I should mention that a very similar work with very similar results has been published in February 2019 in *Pest Management Science*: Simma et al. showed leg cuticle thickening and gene expression changes in insecticide resistant *Anopheles arabiensis*. This article has neither been cited nor discussed by the authors of this work. Of course, the earlier publication of Simma's work questions the novelty of this work.

- 1) Abstract: the effects analysed here are speculated to have an impact on "pheromonal communication" and "resistance to desiccation". To me, this kind of far-fetched speculations doesn't belong to the abstract (see also arguments below).
- 2) The nomenclature for the cuticle regions is not precise enough: According to M. Locke's nomenclature, CHCs are on the envelope, not in the epicuticle.
- 3) Materials & Methods: in the TEM and chitin quantification analyses, the age of the females has not been given.
- 4) Figure 1: Images of resistant and non-resistant tarsi should be shown.
- 5) Most data are shown for females; this should be emphasised throughout the manuscript, i.e. also in the abstract or even in the title.
- 6) Chitin content and cuticle thickness was measured in the legs. If this is a specific response to insecticide application (besides CHCs), the cuticle and the chitin content of the body should not be visibly affected. This has not been shown. Hence, a "leg cloak" has not been shown.
- 7) "Chitin filaments" inferred from D-glucosamine amounts is bold. This is an interpretation, not a direct result.
- 8) Cpr8 and Cpr120 were down-regulated in the legs. Hence, not only cuticle thickness is changed but also cuticle quality. Hypothetically, these two proteins may alone account for the resistance. A possible mechanism can be discussed by the authors. Were these two proteins also found in *A. arabiensis* to be down-regulated?
- 9) The discussion about ABCH transporters is not accurate. Zuber et al., showed that the barrier function of the cuticle is compromised in ABCH deficient *Drosophila* larvae and wings. The envelope fails to be formed in these mutants.

Review form: Reviewer 3

Recommendation

Accept with minor revision (please list in comments)

Scientific importance: Is the manuscript an original and important contribution to its field?

Excellent

General interest: Is the paper of sufficient general interest?

Good

Quality of the paper: Is the overall quality of the paper suitable?

Excellent

Is the length of the paper justified?

Yes

Should the paper be seen by a specialist statistical reviewer?

No

Do you have any concerns about statistical analyses in this paper? If so, please specify them explicitly in your report.

No

It is a condition of publication that authors make their supporting data, code and materials available - either as supplementary material or hosted in an external repository. Please rate, if applicable, the supporting data on the following criteria.

Is it accessible?

Yes

Is it clear?

Yes

Is it adequate?

Yes

Do you have any ethical concerns with this paper?

No

Comments to the Author

The submitted study describes an investigation of the mechanisms underpinning cuticular resistance to insecticides in mosquito disease vectors. The work is comprehensive and provides significant new information on a notoriously difficult form of resistance to investigate. I have only very minor suggestions for improvement prior to publication.

Perhaps one of the native English co-authors could go through the manuscript and make minor changes to smooth the English: 'over-represented in the legs of resistant legs compared to susceptible ones'. Consider also changing the title too - 'Mosquitoes cloak their legs to resist insecticides'

Some sections of the results are very brief, for example section 3 and could benefit from inclusion of more background/general information on the results obtained. For example while there is no

mention of number of proteins identified/analysed in the results this information appears in the discussion.

The upregulation of GSTs in legs of resistant mosquitoes is mentioned in the discussion but not discussed – what might the functional significance of this be?

Decision letter (RSPB-2019-0558.R0)

17-Apr-2019

Dear Professor Vontas:

I am writing to inform you that your manuscript RSPB-2019-0558 entitled "Mosquito build leg cloak to fight against insecticides" has, in its current form, been rejected for publication in Proceedings B.

This action has been taken on the advice of referees, who have recommended that substantial revisions are necessary. With this in mind we would be happy to consider a resubmission, provided the comments of the referees are fully addressed. However please note that this is not a provisional acceptance. Note that convincing reviewers concerning novelty is an important aspect for publication in Proceedings B (see comments by reviewer 2).

Sincerely,

Proceedings B
mailto: proceedingsb@royalsociety.org

Reviewer(s)' Comments to Author:

Referee: 1

Comments to the Author(s)

The authors are to be congratulated for a highly significant innovative study that shows how insecticide resistant mosquitoes structurally and physiologically build a thickened leg cloak in response to insecticide exposure. Research to identify mechanisms is excellent and carefully done. This is a landmark paper that opens a new field of research. No suggestions for improvement, except very minor edits.

Referee: 2

Comments to the Author(s)

In their work "Mosquito build leg cloak to fight against insecticides", Vasileia Balabanidou and colleagues report on the resistance of *Anopheles gambiae* to insecticides through changes in the tarsal cuticle thickness and composition.

Overall, reading of the manuscript was cumbersome; this may be a matter of style though.

Before commenting the data of this work, I should mention that a very similar work with very similar results has been published in February 2019 in *Pest Management Science*: Simma et al. showed leg cuticle thickening and gene expression changes in insecticide resistant *Anopheles arabiensis*. This article has neither been cited nor discussed by the authors of this work. Of course, the earlier publication of Simma's work questions the novelty of this work.

1) Abstract: the effects analysed here are speculated to have an impact on "pheromonal communication" and "resistance to desiccation". To me, this kind of far-fetched speculations doesn't belong to the abstract (see also arguments below).

2) The nomenclature for the cuticle regions is not precise enough: According to M. Locke's nomenclature, CHCs are on the envelope, not in the epicuticle.

3) Materials & Methods: in the TEM and chitin quantification analyses, the age of the females has not been given.

4) Figure 1: Images of resistant and non-resistant tarsi should be shown.

5) Most data are shown for females; this should be emphasised throughout the manuscript, i.e. also in the abstract or even in the title.

6) Chitin content and cuticle thickness was measured in the legs. If this is a specific response to insecticide application (besides CHCs), the cuticle and the chitin content of the body should not be visibly affected. This has not been shown. Hence, a "leg cloak" has not been shown.

7) "Chitin filaments" inferred from D-glucosamine amounts is bold. This is an interpretation, not a direct result.

8) Cpr8 and Cpr120 were down-regulated in the legs. Hence, not only cuticle thickness is changed but also cuticle quality. Hypothetically, these two proteins may alone account for the resistance. A possible mechanism can be discussed by the authors. Were these two proteins also found in *A. arabiensis* to be down-regulated?

9) The discussion about ABCH transporters is not accurate. Zuber et al., showed that the barrier function of the cuticle is compromised in ABCH deficient *Drosophila* larvae and wings. The envelope fails to be formed in these mutants.

Referee: 3

Comments to the Author(s)

The submitted study describes an investigation of the mechanisms underpinning cuticular resistance to insecticides in mosquito disease vectors. The work is comprehensive and provides significant new information on a notoriously difficult form of resistance to investigate. I have only very minor suggestions for improvement prior to publication.

Perhaps one of the native English co-authors could go through the manuscript and make minor changes to smooth the English: 'over-represented in the legs of resistant legs compared to susceptible ones'. Consider also changing the title too - 'Mosquitoes cloak their legs to resist insecticides'

Some sections of the results are very brief, for example section 3 and could benefit from inclusion of more background/general information on the results obtained. For example while there is no mention of number of proteins identified/analysed in the results this information appears in the discussion.

The upregulation of GSTs in legs of resistant mosquitoes is mentioned in the discussion but not discussed - what might the functional significance of this be?

Author's Response to Decision Letter for (RSPB-2019-0558.R0)

See Appendix A.

RSPB-2019-1091.R0

Review form: Reviewer 2

Recommendation

Reject - article is not of sufficient interest (we will consider a transfer to another journal)

Scientific importance: Is the manuscript an original and important contribution to its field?

Good

General interest: Is the paper of sufficient general interest?

Excellent

Quality of the paper: Is the overall quality of the paper suitable?

Excellent

Is the length of the paper justified?

Yes

Should the paper be seen by a specialist statistical reviewer?

No

Do you have any concerns about statistical analyses in this paper? If so, please specify them explicitly in your report.

No

It is a condition of publication that authors make their supporting data, code and materials available - either as supplementary material or hosted in an external repository. Please rate, if applicable, the supporting data on the following criteria.

Is it accessible?

N/A

Is it clear?

N/A

Is it adequate?

N/A

Do you have any ethical concerns with this paper?

No

Comments to the Author

The authors have substantially and convincingly clarified their points. However, after reading even more about the problem, I cannot feel impelled to accept the novelty of this work, which, nevertheless and certainly is the most comprehensive in that field.

Decision letter (RSPB-2019-1091.R0)

24-Jun-2019

Dear Professor Vontas

I am pleased to inform you that your Review manuscript RSPB-2019-1091 entitled "Mosquitoes cloak their legs to resist insecticides" has been accepted for publication in Proceedings B.

The referee and the Associate Editor do not recommend any further changes. Therefore, please proof-read your manuscript carefully and upload your final files for publication. Because the schedule for publication is very tight, it is a condition of publication that you submit the revised version of your manuscript within 7 days. If you do not think you will be able to meet this date please let me know immediately.

To upload your manuscript, log into <http://mc.manuscriptcentral.com/prsb> and enter your Author Centre, where you will find your manuscript title listed under "Manuscripts with Decisions." Under "Actions," click on "Create a Revision." Your manuscript number has been appended to denote a revision.

You will be unable to make your revisions on the originally submitted version of the manuscript. Instead, upload a new version through your Author Centre.

1) A text file of the manuscript (doc, txt, rtf or tex), including the references, tables (including captions) and figure captions. Please remove any tracked changes from the text before submission. PDF files are not an accepted format for the "Main Document".

2) A separate electronic file of each figure (tiff, EPS or print-quality PDF preferred). The format should be produced directly from original creation package, or original software format. Please note that PowerPoint files are not accepted.

3) Electronic supplementary material: this should be contained in a separate file from the main text and the file name should contain the author's name and journal name, e.g. `authorname_procb_ESM_figures.pdf`

All supplementary materials accompanying an accepted article will be treated as in their final form. They will be published alongside the paper on the journal website and posted on the online figshare repository. Files on figshare will be made available approximately one week before the accompanying article so that the supplementary material can be attributed a unique DOI. Please see: <https://royalsociety.org/journals/authors/author-guidelines/>

4) Data-Sharing and data citation

It is a condition of publication that data supporting your paper are made available. Data should be made available either in the electronic supplementary material or through an appropriate repository. Details of how to access data should be included in your paper. Please see <https://royalsociety.org/journals/ethics-policies/data-sharing-mining/> for more details.

<http://datadryad.org/submit?journalID=RSPB&manu=RSPB-2019-1091> which will take you to your unique entry in the Dryad repository.

Once again, thank you for submitting your manuscript to Proceedings B and I look forward to receiving your final version. If you have any questions at all, please do not hesitate to get in touch.

Sincerely,

Professor Hans Heesterbeek

Reviewer(s)' Comments to Author:

Referee: 2

Comments to the Author(s).

The authors have substantially and convincingly clarified their points. However, after reading even more about the problem, I cannot feel impelled to accept the novelty of this work, which, nevertheless and certainly is the most comprehensive in that field.

Sincerely,

Proceedings B

Decision letter (RSPB-2019-1091.R1)

28-Jun-2019

Dear Professor Vontas

I am pleased to inform you that your manuscript entitled "Mosquitoes cloak their legs to resist insecticides" has been accepted for publication in Proceedings B.

Open Access

Paper charges

Sincerely,

Appendix A

Reviewer(s)' Comments to Author and Responses:

Referee: 1

Comments to the Author(s)

The authors are to be congratulated for a highly significant innovative study that shows how insecticide resistant mosquitoes structurally and physiologically build a thickened leg cloak in response to insecticide exposure. Research to identify mechanisms is excellent and carefully done. This is a landmark paper that opens a new field of research. No suggestions for improvement, except very minor edits.

Response:

Thank you very much for your comments.

Referee: 2

Comments to the Author(s)

In their work “Mosquito build leg cloak to fight against insecticides”, Vasileia Balabanidou and colleagues report on the resistance of *Anopheles gambiae* to insecticides through changes in the tarsal cuticle thickness and composition.

Overall, reading of the manuscript was cumbersome; this may be a matter of style though.

Before commenting the data of this work, I should mention that a very similar work with very similar results has been published in February 2019 in *Pest Management Science*: Simma et al. showed leg cuticle thickening and gene expression changes in insecticide resistant *Anopheles arabiensis*. This article has neither been cited nor discussed by the authors of this work. Of course, the earlier publication of Simma’s work questions the novelty of this work.

Response:

We thank Reviewer 2 very much for the excellent comprehensive and detailed review provided.

However, in this particular major criticism for our manuscript, we believe there is a misunderstanding and we do not agree with the Reviewer statement “Of course, the earlier publication of Simma’s work questions the novelty of this work”, as we explain below (and we really hope Reviewer 2 will now see the point – novelty more clear):

Simma et al (2019) was a follow up of an original publication from our team (Balabanidou et al 2016, *PNAS* 113, 9268-9273) which reports cuticle modifications associated with pyrethroid resistance by different mosquito populations (*An arabiensis* in Simma’s case) across Africa.

Like numerous other papers by other authors Simma et al (2019), conducted a genome-wide transcriptome profiling by RNAseq using WHOLE MOSQUITOES and identified differentially expressed transcripts in the resistant population, including detoxification genes and cuticle genes, indeed (similar findings also in several other papers, for example ..).

Simma et al (2019) also analysed cuticle thickness of leg cuticle at the FEMUR as well as CHC in WHOLE MOSQUITO, based literally on Balabanidou et al 2016 experimental.

We did not cite Simma et al (2019) because it did not report any new information compared to the articles that were cited already in our manuscript (i.e.: ... cuticle transcripts in whole mosquitoes; CHC and cuticle thickness in Balabanidou et al 2016) - however, to respond the reviewer comment and make sure there is no misunderstanding, we did so in our revised manuscript (Simma et al 2019 added in the reference list).

In contrast to the paper of Balabanidou 2016 and Simma 2019, the manuscript that we submit in Proceedings B does present novel experimental and findings:

1. An advanced Transmission Electron Microscopy approach was developed to monitor cuticle thickness in the most distal and thin leg segment, the TARSUS (but not possibly less relevant Femur previously studied), that contacts directly insecticide molecules and therefore is the most relevant leg segment to study. We report for the first time that leg Tarsus is thicker in insecticide resistant *Anopheles* mosquitoes compared to susceptible ones.
2. A leg specific CHC determination was introduced for the first time, which showed that CHCs were specifically increased by 43.15 % in the resistant legs (compared to CHC extracted from legs of Susceptible mosquitoes, with the respective difference estimated at only 16.37% when measured in extracts of whole insects, like in all previous studies (Balabanidou et al 2016, Simma et al 2019). More over a leg-specific CHCs signature was identified, possibly implicating a selective transport of CHCs towards the legs.
3. The OMIC analysis reported in our manuscript has been (a) performed at the proteomic level (in contrast to all previous studies which were transcriptomic approaches) and (b) focused on the leg specifically but did not analyse whole bodies (like all previous studies did) to avoid masking effects. It identified CPs, structural protein components of cuticle, as key modulators of leg cuticle (twenty-nine CPs over-represented in the resistant leg proteome· most of them contain a characteristic chitin-binding motif, which binds the polysaccharide chitin to trigger reinforcement

of cuticle layers) as well as two CPs (CPR8 and CPR 120) down-regulated in the resistant legs possibly indicating that qualitative differences could also affect insecticide penetration.

4. The elevated chitin content as a possible biochemical mechanism underpinning cuticle alternations associated with insecticide resistance is finally being reported for the first time in this manuscript.

Specific comments by Reviewer 2

1) Abstract: the effects analysed here are speculated to have an impact on “pheromonal communication” and “resistance to desiccation”. To me, this kind of far-fetched speculations doesn’t belong to the abstract (see also arguments below).

Response:

Thank you. We agree, this is speculative at present with the current dataset. We have omitted this sentence from the abstract.

2) The nomenclature for the cuticle regions is not precise enough: According to M. Locke’s nomenclature, CHCs are on the envelope, not in the epicuticle.

Response:

Thank you, this is a valid point which we tried to further clarify according to the recommendation:

In earlier studies on the insect cuticle there was a good agreement among authors about the layers which compose the insect epicuticle: Wigglesworth V.B (1947) defined (1) an outer epicuticle and (2) an inner epicuticle (or “cuticulin layer”). The same layers were recognized (but differently named) by Locke (1961): the outermost (1) “cuticulin” layer, of unknown composition but rich in lipids and (2): an inner epicuticle named as “protein epicuticle”, consisting of highly tanned proteins associated with abundant lipids. In 2001, Locke proposed a modification in nomenclature incorporating the term "envelope" to replace the cuticulin or “outer epicuticle”. However, it has not been incorporated to a large extent in insect lipid literature, although probably more in ultra-structural studies.

Thus, the "envelope" refers to the "outer epicuticle or cuticulin layer" of both authors.

In the cuticle lipid field, plenty literature has been published before (and after) Locke’s envelope proposal. It is widely accepted that the so called epicuticle lipid layer is located on the outermost surface of the epicuticle; the “envelope” term has not been incorporated yet to an evident extent.

This surface layer is easily washed with solvent, and there is plenty literature showing how solvent easily extract surface lipid deposits. This knowledge was developed after seminal studies on cuticle structure and composition by Dr Wigglesworth and also Dr Locke (among others: J. Cell Set. 19, 459-485 (1975), Science Vol 147, pp 295-298, 1965).

To improve clarity, based on the pertinent reviewer recommendation, we have added a sentence in introduction (lines 76-77): "Epicuticle surface is covered by lipids, the majority of which are hydrocarbons (CHCs), free fatty acids and wax esters, whereas internally highly tanned proteins associated with lipids are located".

3) Materials & Methods: in the TEM and chitin quantification analyses, the age of the females has not been given.

Response:

Apologies. We have added the age of the mosquitoes (3-5 days old mosquitoes).

4) Figure 1: Images of resistant and non-resistant tarsi should be shown.

Response:

Thank you. Both images have been added in the revised manuscript.

5) Most data are shown for females; this should be emphasized throughout the manuscript, i.e. also in the abstract or even in the title.

Response:

Absolutely correct. It has been emphasized in the abstract (lines 33, 36) and in the text and in several places.

6) Chitin content and cuticle thickness was measured in the legs. If this is a specific response to insecticide application (besides CHCs), the cuticle and the chitin content of the body should not be visibly affected. This has not been shown. Hence, a "leg cloak" has not been shown.

Response:

Thank you, this is also a valid point.

A substantially larger amount of D-glucosamine (58.5% more) was found to be present in the leg cuticle of insecticide resistant mosquitoes (0.71 mgr of D-glucosamine/ mgr of mosquito legs \pm 0.03) compared to the susceptible mosquitoes (0.3 mgr of D-glucosamine/ mgr of mosquito legs \pm 0.02) ($P < 0.0001$). This difference was diminished when glucosamine from body extracts was measured in both resistant (1.46

mgr of D-glucosamine/ mgr of mosquito bodies ± 0.49) and susceptible mosquitoes (1.12 mgr of D-glucosamine/ mgr of mosquito bodies ± 0.16). This new information has been incorporated in the supplement (see Supplement and supplementary Figure S5) and discussed in lines 338-341 in the revised manuscript.

A comprehensive analysis on cuticle from different body regions was not attempted, as the standardization of those measurements on different parts of the body (possibly irrelevant to our study) is not trivial.

7) “Chitin filaments” inferred from D-glucosamine amounts is bold. This is an interpretation, not a direct result.

Response:

Thank you, corrected: chitin filaments has been replaced with D-glucosamine or chitin content throughout the manuscript.

8) Cpr8 and Cpr120 were down-regulated in the legs. Hence, not only cuticle thickness is changed but also cuticle quality. Hypothetically, these two proteins may alone account for the resistance. A possible mechanism can be discussed by the authors. Were these two proteins also found in *A. arabiensis* to be down-regulated?

Response:

Thank you, we agree, this is another valid point.

One possible explanation could be the involvement of these structural components in pore canal formation that facilitate insecticide penetration. However, we have no direct evidence for it, at present. Nevertheless, we have added a brief discussion of this point (lines 411-413).

Anopheles gambiae CPR120 does not have a known orthologue to *Anopheles arabiensis*. CPR8 orthologue in *An. arabiensis* is AARA004240. AGAP004240 (pointed with an arrow in the heatmap below) was not found to be differentially regulated in three resistant strains versus the susceptible tested while one out of the four resistant strains tested shows a slight up-regulation as opposed to the susceptible (Simm et al., 2019).

Expression heatmap of cuticle related genes of *An. arabiensis*. Cuticle related genes were defined as those genes coding for proteins with one of the following InterPro domains: IPR000618, IPR031311, IPR31874, IPR002557, IPR22727, IPR004302 or IPR004835. The log₂-transformed gene fold changes of the Ethiopian deltamethrin/DDT resistant populations ASN, CHW, TOL and the susceptible strain MOZ from Mozambique are relative to the susceptible SEK strain from Ethiopia. Genes without expression values in all four comparisons were excluded from the heatmap. *Anopheles arabiensis* gene IDs are shown on the right (Simma et al., 2019).

9) The discussion about ABCH transporters is not accurate. Zuber et al., showed that the barrier function of the cuticle is compromised in ABCH deficient *Drosophila* larvae and wings. The envelope fails to be formed in these mutants.

Response:

Thank you. Zuber *et al* (2018) showed that *Dm_snu* is responsible for the formation of a lipid-based cuticle barrier, since silencing of the respective gene resulted in increased permeability of lipophilic dyes, such as Eosin Y, at 50° C. Lipophilic components in or at the cuticle are depleted when *snu* is dysfunctional. A phylogenetic analysis presented below confirmed that AGAP003680-PA is an ortholog of *Snu* gene. This work, cited now in the revised manuscript, together with the studies in *Tribolium castaneum* (Broehan G, et al., 2013) and *Locusta migratoria* (Yu, et al., 2017), seems to support our hypothesis for a possible implication of AGAP003680-PA in lipid-transport towards the cuticle, and we hope Reviewer 2 is agreeable to please leave it.

Tree representing phylogenetic relationship among ABCH transporters from *Anopheles gambiae*, *Tribolium castaneum*, *Drosophila melanogaster* and *Helicoverpa armigera*.

Referee: 3

Comments to the Author(s)

The submitted study describes an investigation of the mechanisms underpinning cuticular resistance to insecticides in mosquito disease vectors. The work is comprehensive and provides significant new information on a notoriously difficult form of resistance to investigate.

Thank you.

I have only very minor suggestions for improvement prior to publication.

Perhaps one of the native English co-authors could go through the manuscript and make minor changes to smooth the English: 'over-represented in the legs of resistant legs compared to susceptible ones'. Consider also changing the title too – 'Mosquitoes cloak their legs to resist insecticides'

Response:

Thank you very much - the title has been revised according to the excellent recommendation!

Some sections of the results are very brief, for example section 3 and could benefit from inclusion of more background/general information on the results obtained. For example while there is no mention of number of proteins identified/analysed in the results this information appears in the discussion.

Response:

Thank you. A more detailed description of the proteomic results is presented in the supplementary material (ESM).

The upregulation of GSTs in legs of resistant mosquitoes is mentioned in the discussion but not discussed – what might the functional significance of this be?

Response:

Thank you. We have discussed the possible functional significance in lines 426-436 in the revised manuscript.